# A growth selection system for the directed evolution of amine-forming or converting enzymes

Shuke Wu [1,2,6] ✉, Chao Xiang[1,6], Yi Zhou[2], Mohammad Saiful Hasan Khan [1], Weidong Liu [1,3], Christian G. Feiler[4], Ren Wei [1], Gert Weber [4], Matthias Höhne [5] & Uwe T. Bornscheuer [1] ✉

Fast screening of enzyme variants is crucial for tailoring biocatalysts for the asymmetric synthesis of non-natural chiral chemicals, such as amines. However, most existing screening methods either are limited by the throughput or require specialized equipment. Herein, we report a simple, high-throughput, low-equipment dependent, and generally applicable growth selection system for engineering amine-forming or converting enzymes and apply it to improve biocatalysts belonging to three different enzyme classes. This results in (i) an amine transaminase variant with 110-fold increased specific activity for the asymmetric synthesis of the chiral amine intermediate of Linagliptin; (ii) a 270-fold improved monoamine oxidase to prepare the chiral amine intermediate of Cinacalcet by deracemization; and (iii) an ammonia lyase variant with a 26-fold increased activity in the asymmetric synthesis of a non-natural amino acid. Our growth selection system is adaptable to different enzyme classes, varying levels of enzyme activities, and thus a flexible tool for various stages of an engineering campaign.

Chiral amines are indispensable building blocks for a large variety of bioactive pharmaceuticals and agrochemicals, and they are also widely employed as chiral auxiliaries and resolving agents. Consequently, catalytic synthesis of optically pure amines has been the key subject in chemical and pharmaceutical industries[1–4]. In this context, biocatalytic approaches are very attractive for highly selective and green synthesis of chiral chemicals[5–10]. Recent discoveries and developments of transaminases[11–15], monoamine oxidases[16], amine dehydrogenases[17], imine reductases[18], and ammonia lyases[19] provide a versatile biocatalyst toolbox to access chiral amines. Since many synthetically useful chiral intermediates are new to these natural enzymes, extensive and time-consuming directed evolution[20–23] is usually required to tailor

natural enzymes for the production of specific non-natural chiral amine targets[24–26]. Therefore, high-throughput screening methods have been developed for specific classes of enzymes[27,28]. For example, specialized amine donors that elicit optical responses were exploited for assaying transaminases[29]; monoamine oxidase activities are visualized with $H_2O_2$-coupled assays[30]; and NAD(P)⁺/NAD(P)H-coupled assays facilitate screening of dehydrogenases/reductases variants[31]. For challenging lyase-catalyzed reactions without easily detectable co-products, mutant design mostly had to rely on computational redesign[32,33] or expensive high-throughput mass spectrometry[34]. However, most of these methods either are still limited by the throughput (when applied in multi-well plates) or require specialized

[1]Department of Biotechnology and Enzyme Catalysis, Institute of Biochemistry, University of Greifswald, Felix Hausdorff-Str. 4, D–17489 Greifswald, Germany. [2]State Key Laboratory of Agricultural Microbiology, College of Life Science and Technology, Huazhong Agricultural University, No. 1 Shizishan Street, Wuhan 430070, PR China. [3]Industrial Enzymes National Engineering Laboratory, Tianjin Institute of Industrial Biotechnology, Chinese Academy of Sciences, Tianjin 300308, PR China. [4]Macromolecular Crystallography, Helmholtz-Zentrum Berlin für Materialien und Energie, Albert-Einstein-Straße 15, 12489 Berlin, Germany. [5]Protein Biochemistry, Institute of Biochemistry, University of Greifswald, Felix Hausdorff-Str. 4, D-17489 Greifswald, Germany. [6]These authors contributed equally: Shuke Wu, Chao Xiang. ✉e-mail: shukewu@mail.hzau.edu.cn; uwe.bornscheuer@uni-greifswald.de

and expensive equipment (e.g., robotic platforms[35] or microfluidic devices[30,36]), thus preventing a more widespread implementation of these enzymes in biocatalysis. Furthermore, none of these methods is applicable for different classes of enzymes. Therefore, a high-throughput, low-equipment dependent, and generally applicable concept is highly desired for the directed evolution of enzymes for the synthesis of non-natural chiral amines.

In contrast to screening with the methods mentioned above, growth selection is intrinsically connected with high-throughput and simplicity because only the desired highly active variants (instead of every variant) of the library are detectable and generate cell growth as an easily measurable output signal via the optical density (OD) in liquid culture or colony formation/size in solid culture[37–39]. However, selection requires a direct link between the activity of the enzyme and the survival of the host, which is often highly specialized and difficult to be established for many synthetically useful enzymes. Pioneering studies in selection-based enzyme evolution often focused on enzymes conferring antibiotic resistance (e.g., β-lactamases)[40,41] targeting proteins linked to the expression of antibiotic resistance genes[42] or phage coat protein pIII (essential for phage propagation)[43]. Yet, it is difficult to apply this concept to synthetically useful enzymes. Another approach is to exploit the auxotrophy generated by deleting essential genes for certain metabolites (e.g., natural amino acids)[44–46] and then to complement them by directed evolution of related enzymes, which are usually limited to those cases where natural metabolites or biochemicals are produced as exemplified for a chorismate mutase[45] or natural amino acid racemases[46]. Very recently, redox cofactor (NAD(P)+/NAD(P)H) auxotrophs were developed via extensive genome engineering, and employed for the selection-based evolution of several NAD(P)H-dependent enzymes[47–50]. This approach has proven useful in switching the cofactor preference of several enzymes (e.g., formate dehydrogenase)[47,48], yet with limited success in improving activities (~10-fold) for non-native substrates[49,50]. Despite the fact that growth selection is theoretically a very powerful strategy, its great potential of evolving enzymes for asymmetric synthesis/kinetic resolution is underrepresented, except for early reports on the selection-based evolution of hydrolases[51,52]. There is rare report of a selection-based improvement of enzyme activities for the synthesis of non-natural chiral amines.

Herein, we report a simple, high-throughput, low-equipment dependent, and generally applicable growth selection system for the directed evolution of amine-forming or converting enzymes (Fig. 1). This methodology is demonstrated by the evolution of three different classes of enzymes for the synthesis of three important non-natural chiral amines in high enantiopurity: (1) an amine transaminase (TA, enzyme class EC 2) for (R)-1-Boc-3-aminopiperidine, the key chiral intermediate for the antidiabetic drugs Linagliptin, Trelagliptin and Alogliptin; (2) a monoamine oxidase (MAO, enzyme class EC 1) for (R)-

1-(1-naphthyl)ethylamine, the key intermediate for the calcimimetic drug Cinacalcet; and (3) an ammonia lyase (enzyme class EC 4) for (S)-2-amino-3-(naphthalen-1-yl)propanoic acid, the key chiral synthon for a highly potent and long-acting Gα12 agonist. The rapid improvements of catalytic activities (26–270-fold) in only one or two rounds of evolution without specialized equipment showcase the great potential of a growth selection system for wider application for the creation of suitable biocatalysts to meet the speed requirement in industry.

## Results
### Design and general procedure of the growth selection system
To design a generally applicable growth selection system for amine-forming or converting enzymes, we exploited growth based on alanine or ammonia formation generated from the targeted amine by amine-forming or converting enzymes: reversible transamination of the targeted amine by a TA with intracellular pyruvate releases L- or D-alanine (Fig. 1a); oxidation of the targeted amine by the MAO produces the imine, which is auto-hydrolyzed to release ammonia (Fig. 1b); the reversible conversion of the targeted amino acid by the lyase produces an α,β-unsaturated acid and ammonia (Fig. 1c). Alanine (L- or D-) or ammonia can then be easily utilized by the Escherichia coli strain as the nitrogen source for cell growth (Fig. 1d). Hence, when the targeted amine is supplied as the only nitrogen source in a chemically-defined medium, only the cells containing active enzyme variants (producing alanine or ammonia) can survive and grow. The precondition is that the host cannot directly utilize the targeted amines. Fortunately, this is often valid for many synthetically useful non-natural amines, because E. coli BL21(DE3) has a rather limited scope of nitrogen sources (Supplementary Fig. 1) and its endogenous transaminases are mainly acting on amino acids. Indeed, the use of non-natural amines as the only nitrogen source was often applied to isolate wild-type microorganisms containing amine-converting enzymes. However, this strategy has rarely been demonstrated for the directed evolution of enzymes for the synthesis of non-natural chiral amines.

The general procedure of the growth-based directed evolution system to improve amine-forming or converting enzymes is proposed as follows (Fig. 2): (1) genetic construction of a vector harboring the gene encoding the enzyme of interest with a suitable promoter, and mutagenesis by PCR (e.g., multiple site-directed mutagenesis, error-prone PCR or DNA shuffling, Fig. 2a); (2) highly efficient transformation of E. coli with the constructed gene library by electroporation (Fig. 2b); (3) selection of improved variants within the E. coli library on the M9 plate or liquid medium (enrichment followed by spreading on LB agar plates) with the targeted amine as the only nitrogen source (Fig. 2c); (4) isolation of the single colonies, sequencing of the genes, purification, and characterization of the enzyme variants (Fig. 2d). By this, a genotype-phenotype linkage is ensured. A common issue in the growth selection is that the cell growth rate may not be proportional to

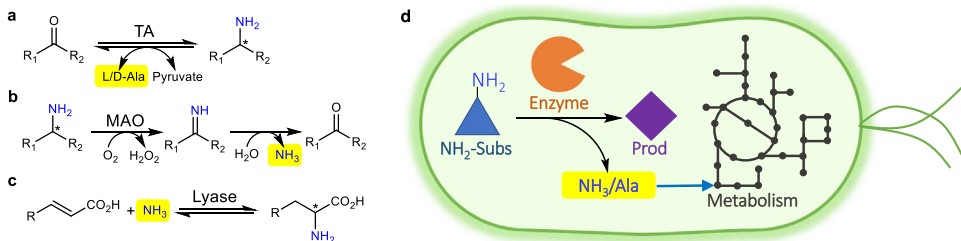

**Fig. 1 | Concept of the growth selection system for the directed evolution of transaminases (TAs), monoamine oxidases (MAOs), and ammonia lyases. a** The TA catalyzes the reversible transamination of a targeted amine and releases easily usable L- or D-alanine (highlighted). **b** The MAO catalyzes oxidation of a targeted amine to its imine which is auto-hydrolyzed to release ammonia (highlighted). **c** The lyase catalyzes the reversible conversion of an α,β-unsaturated acid and ammonia (highlighted) to a targeted amino acid. **d** Alanine or ammonia generated from a targeted amine by the active enzymes is then utilized in situ for cell growth serving as an output signal.

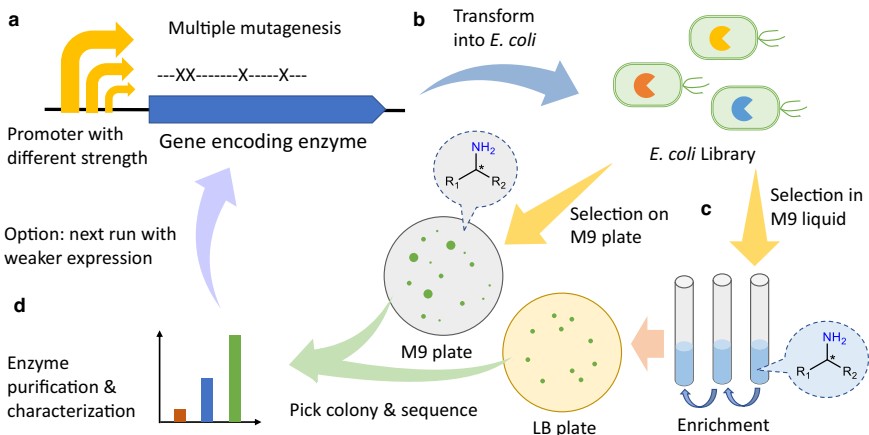

**Fig. 2 | General procedure of the growth-based directed evolution system to improve amine-forming or converting enzymes. a** Mutagenesis of the gene encoding the enzyme of interest cloned under a promoter with suitable strength. **b** *E. coli* host cells are transformed with the gene library. **c** Selection with M9 plates or liquid medium with the targeted amine as the only nitrogen source. **d** Isolation of single colonies, sequencing of the genes, and characterization of the enzyme variants.

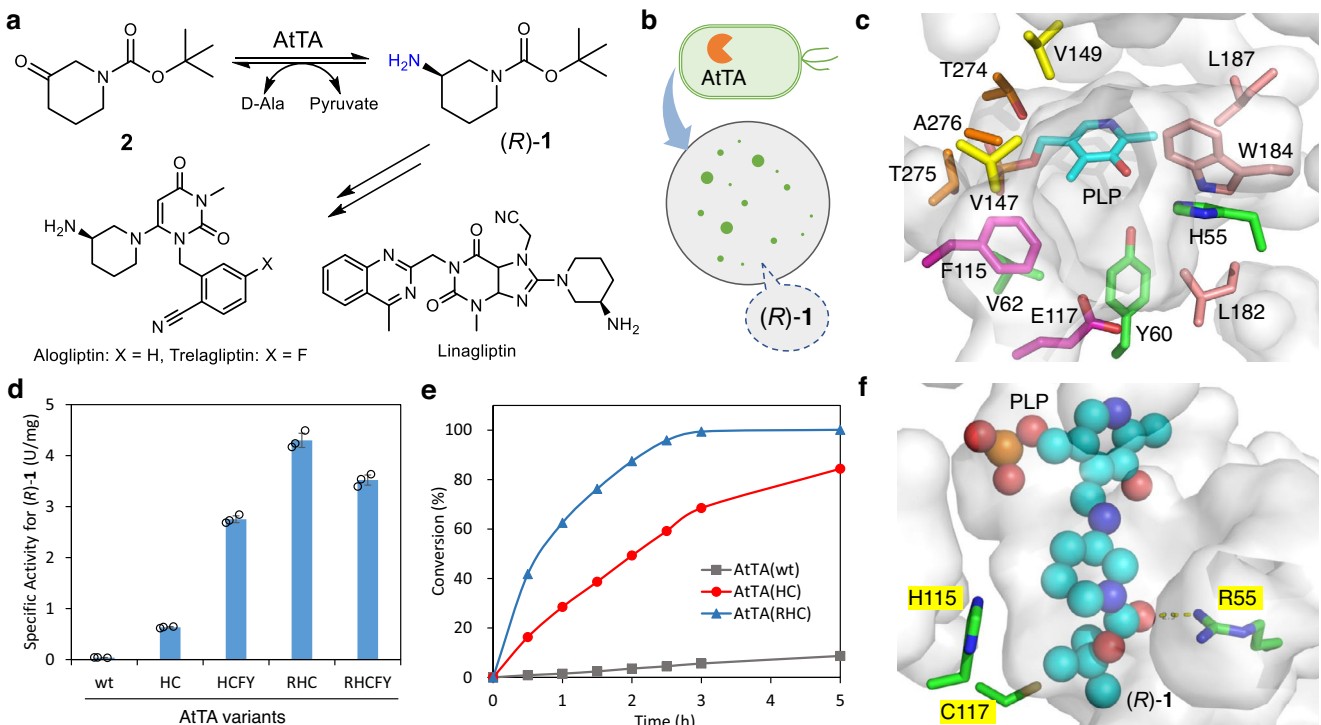

**Fig. 3 | Growth-based directed evolution of AtTA for the asymmetric synthesis of (*R*)-1. a** Targeted reaction converting **2** to (*R*)-**1** in an asymmetric synthesis. **b** The principle of the growth selection is based on M9 agar plate selection where the amine (*R*)-**1** serves as the only nitrogen source. **c** Active site of AtTA(wt) (PDB: 4CE5) showing the target sites (grouped in different colors) for mutagenesis. **d** Specific activity of purified AtTA variants in the conversion of (*R*)-**1** with pyruvate. **e** Time course of the asymmetric synthesis of (*R*)-**1** from **2** by purified AtTA variants. **f** Docking of the PLP-(*R*)-**1** complex (represented as cyan spheres) in the active site of the structure of AtTA(RHC) containing the mutations H55R/F115H/E117C. AtTA: amine transaminase from *Aspergillus terreus*; PLP: pyridoxal-5'-phosphate. Source data are provided as a Source Data file. Data in **d** are mean values of triplicate experiments with error bars indicating the s.d. (*n* = 3). Data in **e** are from one independent experiment.

the specific activity of the enzyme due to the complexity of the cell growth. To address this, we employed four constitutive promoters with different strengths (Supplementary Fig. 2) on the vectors to modulate the expression levels of the enzyme to fine-tune the selection pressure. A strong promoter resulting in high expression will facilitate the conversion of sufficient amounts of amine to promote growth when the starting activity is low. A medium or low expression, on the contrary, ensures growth of cells bearing only the most active variants if a moderate or decent template activity already exists. Other

strategies, such as inducible promoter, 5'-untranslated region[41], or protein degradation tags[45] could also be used to fine-tune the expression levels of the enzyme.

## Growth selection-based evolution of an amine transaminase

To demonstrate the functionality and advantages of the growth selection system for the evolution of TAs, we selected the (*R*)-selective AtTA from *Aspergillus terreus* for the synthesis of (*R*)-**1**, the key chiral synthon for Linagliptin[53], Trelagliptin, and Alogliptin (Fig. 3a)[54].

Although the synthesis of (R)-**1** with a commercial TA was reported recently[55], the sequence of that enzyme is unknown. We previously discovered that AtTA(wt) could be applied for the synthesis of (R)-**1**[56], however, the conversion was only 11%. We found the specific activity of AtTA(wt) towards (R)-**1** is only 0.038 U/mg, which is almost two orders of magnitude lower than for the benchmark substrate (R)-1-phenylethylamine (2.9 U/mg). The efficient synthesis of (R)-**1** by AtTA is thus not possible unless protein engineering significantly improves the specific activity. To verify the feasibility of growth selection, we cloned the AtTA(wt) gene under the control of four promoters with different strengths, and assayed the growth of the resulting transformed *E. coli* cells on M9 agar plates with D-alanine (positive control), (R)-1-phenylethylamine (positive control), and (R)-**1** as the only nitrogen source (Supplementary Fig. 3).

The results show that (1) all *E. coli* could easily use D-alanine for growth; (2) expression of AtTA(wt) at strong to weak levels enabled *E. coli* to grow on (R)-1-phenylethylamine, while *E. coli* was unable to grow with very weak expression of AtTA(wt) or without it (empty vector); (3) all *E. coli* cells were unable to grow on (R)-**1**. Therefore, if the activity of AtTA towards (R)-**1** is remarkably improved, the *E. coli* cell containing this variant should stand out by forming a colony on the plate (Fig. 3b).

To maximize the chance of success in the directed evolution rounds, we first analyzed the structure of AtTA(wt) (PDB: 4CE5)[57] and selected 13 residues in the active site region (Fig. 3c). These residues were grouped into five clusters for the ease of simultaneous mutagenesis: H55-Y60-V62, F115-E117, V147-V149, L182-W184-L187, and T274-T275-A276. Saturation mutagenesis of each group was performed with two or three NNK codons, thus leading to two libraries of 1024 and three libraries of 32,768 unique combinations. The AtTA libraries were constructed using the vectors bearing a strong or medium promoter. These were introduced into *E. coli* BL21(DE3) cells by electroporation, and selected on M9 agar plates with (R)-**1** as the sole nitrogen source. Many colonies appeared on all five plates of AtTA libraries with a strong promoter (Supplementary Fig. 4), indicating that already a slight improvement of activity or expression may allow *E. coli* to grow. On the other hand, medium expressed AtTA allows a more stringent selection: only around 20 colonies were found in the F115-E117 library (Supplementary Fig. 5). The plasmids of 12 representative colonies were isolated to yield eight unique variants, which turned out to have much higher specific activity than the wild-type (0.11–0.53 vs 0.038 U/mg, Supplementary Fig. 6). The best three variants (0.40–0.53 U/mg) were chosen as templates for further mutagenesis. To increase and fine-tune the selection pressure, they were subcloned to the vectors with a weak or very weak promoter. Growth experiments confirmed (Supplementary Fig. 7) that the very weak promoter yields the best condition for the three AtTA variants for further selection and thus we started the second round of evolution by introducing saturation mutagenesis at the other four residue groups. Only the AtTA(YHC) with D5Y/F115H/E117C mutations gave some colonies for all four groups (Supplementary Fig. 8). A total of 96 colonies were sequenced and led to 15 different variants, which were evaluated to yield two significantly more active variants (additional H55R and V147F/V149Y, 1.7–1.9 U/mg, Supplementary Fig. 9). We further combined the beneficial mutants and removed the accidentally introduced D5Y mutation from the first round of evolution. The resulting variants were evaluated (Fig. 3d) and AtTA(RHC) with H55R/F115H/E117C gave the highest activity of 4.2 U/mg (110-fold over the wild type) while maintaining very high enantioselectivity. The best AtTA variants were applied for the asymmetric synthesis of (R)-**1** (Fig. 3e): AtTA(RHC) produced (R)-**1** (98% *ee*) in quantitative conversion in 5 h, while AtTA(wt) only gave (R)-**1** with 8.7% conversion under the same conditions. The preparative scale synthesis was performed with purified AtTA(RHC) to obtain (R)-**1** (98% *ee*) in 98% isolated yield (Supplementary Fig. 10). AtTA(RHC) is one of the best TAs for the synthesis of (R)-**1**.

To elucidate the possible molecular basis of the improved activity (considering the changes in charge, polarity, and size present in the triple mutant H55R/F115H/E117C), we solved the crystal structure of AtTA(RHC) (Supplementary Fig. 11, Supplementary Table 1) with bound PLP (7XG5) and with bound LLP (PLP-Lys180, 7XG6) and performed a docking experiment with the reaction intermediate, the PLP-(R)-**1** complex (the external aldimine). As depicted in Fig. 3f and Supplementary Fig. 12, the PLP part binds to AtTA(RHC) in a very similar position to PLP in the AtTA(wt), and the (R)-**1** part fits into the active site of AtTA(RHC) very well. According to the docking results, particularly the F115H mutation precisely carves out additional space in the substrate-binding pocket to accommodate the piperidine ring of (R)-**1**. E117C expands the pocket and may facilitate the binding of the *tert*-butyl group of (R)-**1**. Furthermore, the H55R variation offers a positively charged side chain which provides a hydrogen bond (2.9 Å) between the guanidinium group and the carbonyl group of (R)-**1**. Therefore, only three precise mutations have been sufficient to significantly (>100-fold) improve the activity of AtTA towards (R)-**1**, enabling practical asymmetric synthesis of this key chiral amine intermediate of several drugs.

## Growth selection-based evolution of a monoamine oxidase

To prove the applicability of growth selection for MAOs, we selected the cyclohexylamine oxidase (CHAO) from *Brevibacterium oxydans* for the deracemization of **3** via simultaneous biocatalytic oxidation and chemical reduction to produce (R)-**3**, the key chiral amine intermediate for the calcimimetic drug Cinacalcet (Fig. 4a)[58]. Although (R)-**3** was prepared by biocatalysis before (mainly via TAs)[59], there are no reports dealing with a biocatalytic deracemization. The specific activities of CHAO(wt) towards (S)- and (R)-**3** were 0.009 and <0.001 U/mg, respectively (Fig. 4d). This indicated that the CHAO(wt) has the desired selectivity towards **3**, but the specific activity towards (S)-**3** is too low for synthetic applications. To degrade the highly toxic $H_2O_2$ generated by CHAO, catalase was co-expressed with CHAO during the selection. Co-expressing of CHAO(wt) at different levels and catalase enabled *E. coli* to grow in M9 liquid culture with ammonia, cyclohexylamine (activity of 5.6 U/mg)[60], or cyclopentylamine (activity of 0.33 U/mg)[60] as the only nitrogen source (Supplementary Fig. 13). Overexpression of the catalase is necessary for growth on substrates, such as cyclohexylamine, which are converted with high activity (Supplementary Fig. 14). However, (S)-**3** and the corresponding ketone are toxic to *E. coli* cells (Supplementary Fig. 15). At a lowered concentration (1 mM) of (S)-**3**, no toxicity effects were observed, but colony formation on agar plate was difficult to evaluate. Therefore, we performed enrichment of *E. coli* cells in liquid cultures and added methyl laurate as a growth-compatible second phase which acts as a reservoir of (S)-**3** and also extracts the ketone in situ. At this reduced concentration of (S)-**3** in the medium, *E. coli* cells bearing active variants of CHAO could be enriched and isolated (Fig. 4b).

To accelerate the developing timelines[61], we envisioned that the growth selection could allow dramatic improvements of activity via a single round of evolution. CHAO(wt) showed a high activity of 3.5 U/mg towards (S)-1-phenylethylamine[60], a similar but less bulky aromatic amine. Thus, it was docked into the active site of CHAO(wt) (PDB: 4I59)[62] to identify possible key residues for mutagenesis (Fig. 4c). We speculated that (S)-**3** may bind in a similar pose but the additional ring of (S)-**3** may occupy either side (F88-F351-L353-F368 or T198-L199-M226-Y321 labeled in salmon or green, Fig. 4c). These two groups of hot spots were subjected to simultaneous mutagenesis using the DBS codon, which stands for 18 different codons encoding 12 hydrophobic and small amino acids: A, R, C, G, I, L, M, F, S, T, W and V. Simultaneously, a four-site mutagenesis of CHAO(wt) was performed with the GoldenGate method[63], leading to two libraries of 104,976 unique combinations. These libraries were constructed using the plasmids with medium to very weak promoters, transformed into *E. coli* BL21(DE3) cells

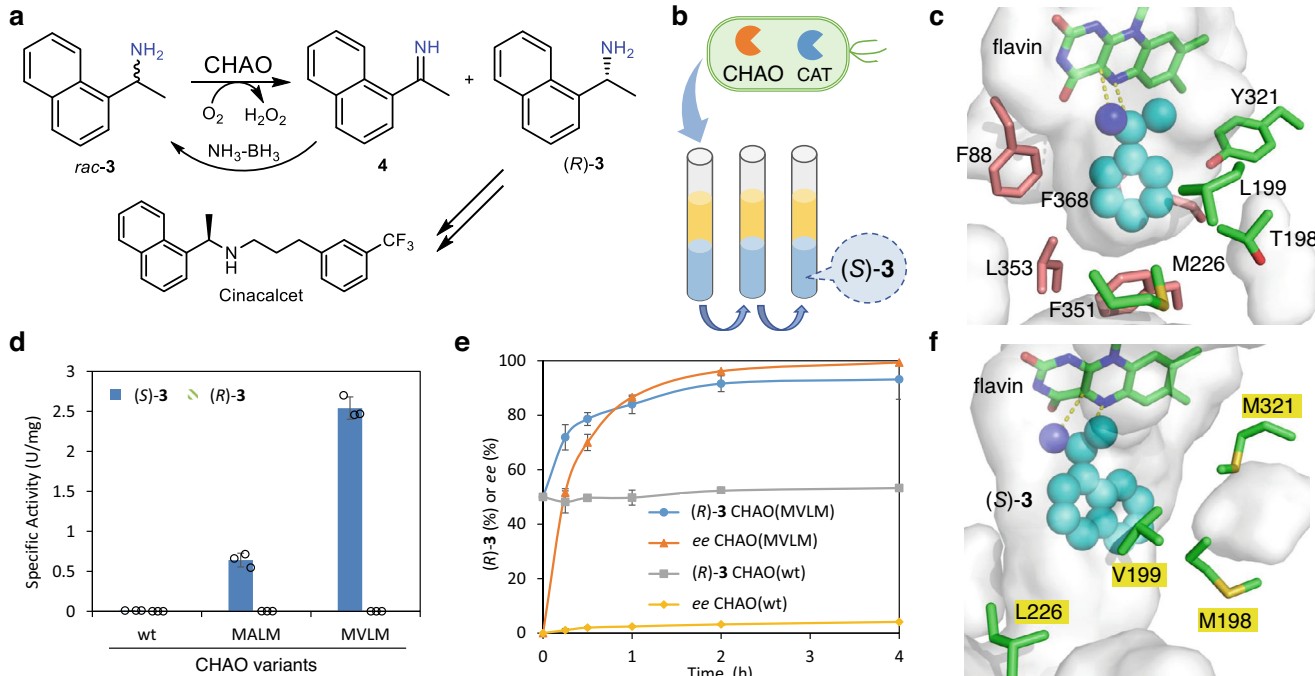

**Fig. 4 | Growth-based directed evolution of CHAO for the deracemization to produce (R)-3. a** Targeted reaction for the deracemization of *rac*-3 to (R)-3. **b** The principle of the growth selection is based on *E. coli* cells co-expressing CHAO variants and a catalase (CAT) in the two-phase system (methyl laurate and M9 liquid medium) where amine (S)-3 (1 mM in the aqueous phase) serves as the only nitrogen source. This ensures a non-toxic concentration but sufficient supply of (S)-3 to the cells and simultaneous removal of the toxic ketone product. **c** Identification of two groups of hot spots by docking of (S)-1-phenylethylamine (truncated substrate, represented as cyan spheres) into the active site of CHAO(wt) (PDB: 4I59). **d** Specific activities of purified CHAO variants for converting (S)- and (R)-3. **e** Time course of the deracemization of *rac*-3 by *E. coli* co-expressing CHAO variants and a catalase in the presence of NH₃-BH₃. **f** Docking of (S)-3 (represented as cyan spheres) in the active site of the structure model of CHAO(MVLM) with T198M/L199V/M226L/Y321M. CHAO: cyclohexylamine oxidase from *Brevibacterium oxydans*; CAT: catalase from *E. coli*. Source data are provided as a Source Data file. Data in **d** and **e** are mean values of triplicate experiments with error bars indicating the s.d. ($n = 3$).

(expressing the catalase) by electroporation, and then underwent enrichment in M9 medium with (S)-3 in the presence of methyl laurate. The culture was diluted every 24 h 5 times, and significant growth was only observed for the medium expressed library of T198-L199-M226-Y321 (Supplementary Fig. 16). The resulting enriched culture was spread on an LB plate, and 12 colonies were isolated to yield two unique variants, CHAO(MALM) with T198M/L199A/M226L/Y321M and CHAO(MVLM) with T198M/L199V/M226L/Y321M mutations. The specific activities of the purified CHAO(MALM) and CHAO(MVLM) towards (S)-3 were 0.64 and 2.54 U/mg, respectively (Fig. 4d). This demonstrated a 70–270-fold increase in specific activities, while the high enantioselectivity was preserved (≤0.001 U/mg towards (R)-3 for both variants). The enzyme kinetics of CHAO(MVLM) towards (S)-3 were $k_{cat}$ = 240 min⁻¹, $K_m$ = 2.2 mM, $K_i$ = 8.4 mM (Supplementary Fig. 17). In comparison with CHAO(wt) ($k_{cat}$ = 0.91 min⁻¹, $K_m$ = 4.5 mM, $K_i$ = 25 mM), the quadruple mutant mainly improved the $k_{cat}$, reduced the $K_m$ but also exhibited low substrate inhibition. *E. coli* cells co-expressing CHAO(MVLM) and catalase were employed for the deracemization of *rac*-3, leading to enantiopure (R)-3 (>99% *ee*) in 93% conversion, while the same reaction with *E. coli* cells expressing CHAO(wt) gave almost racemic 3 (4% *ee*) (Fig. 4e, Supplementary Fig. 18). The preparative scale synthesis was also performed to obtain (R)-3 (>99% *ee*) in 69% isolated yield (Supplementary Fig. 19). As far as we known, CHAO(MVLM) is the best enzyme for the production of (R)-3 by deracemization.

Next, (S)-3 was docked to a homology model of CHAO(MVLM) to elucidate the possible rationale for the observed improved activity (Fig. 4f). According to the docking results, (S)-3 binds to the active site of CHAO(MVLM) in a similar pose but at a slightly different angle compared to (S)-1-phenylethylamine in the CHAO(wt). The Y321M mutation slightly increases the hydrophobicity and size of the pocket. T198M also slightly expands the space by pointing the methionine side chain away from the active site. L199V may provide a suitable hydrophobic interaction with the naphthalene ring. Besides the changes in the active site, the M226L mutation flips the side chain away from F351 and opens the active site to the substrate channel. With proper structure-guided selection of mutation residues, a single round of evolution with growth selection was sufficient to significantly (>200-fold) improve the activity of CHAO towards (S)-3.

**Growth selection-based evolution of an ammonia lyase**

To further demonstrate the applicability of the growth selection system, we have chosen the phenylalanine ammonia lyase from *Petroselinum crispum* (PcPAL)[19] for the synthesis of the non-natural amino acid, (S)-5, the key chiral intermediate for a highly potent Gα12 agonist (Fig. 5a)[64]. The specific activities of PcPAL(wt) towards ʟ-phenylalanine and the target (S)-5 were 0.147 and 0.0027 U/mg, respectively (Fig. 5d), suggesting a large room for activity improvement. However, the solubility of (S)-5 in the culture medium is very low (i.e., <1.5 mM). Furthermore, the growth selection using amino acids is complicated by the presence of indigenous enzymes in the host (*e.g.*, amino acid transaminases). Thus, we tailored the *E. coli* host by expressing the aromatic amino acid transporter (AroP, to increase the availability of (S)-5) and deleting the gene encoding the aromatic amino acid transaminase (*tyrB*, to minimize the interference of indigenous enzymes). We tested the *E. coli* host (with or without deleting *tyrB*) for the expression of PcPAL(wt) at different levels (with or without co-expressing AroP) on M9 agar plates supplemented with ammonia, ʟ-Phe, or (S)-5 as the only nitrogen sources (Supplementary Fig. 20). As expected, deleting *tyrB* significantly reduced the background growth on ʟ-Phe, while expressing AroP contributed to the fast utilization of ʟ-Phe. With these two modifications, there was a better correlation between growth and promoter strength. Although no obvious cell

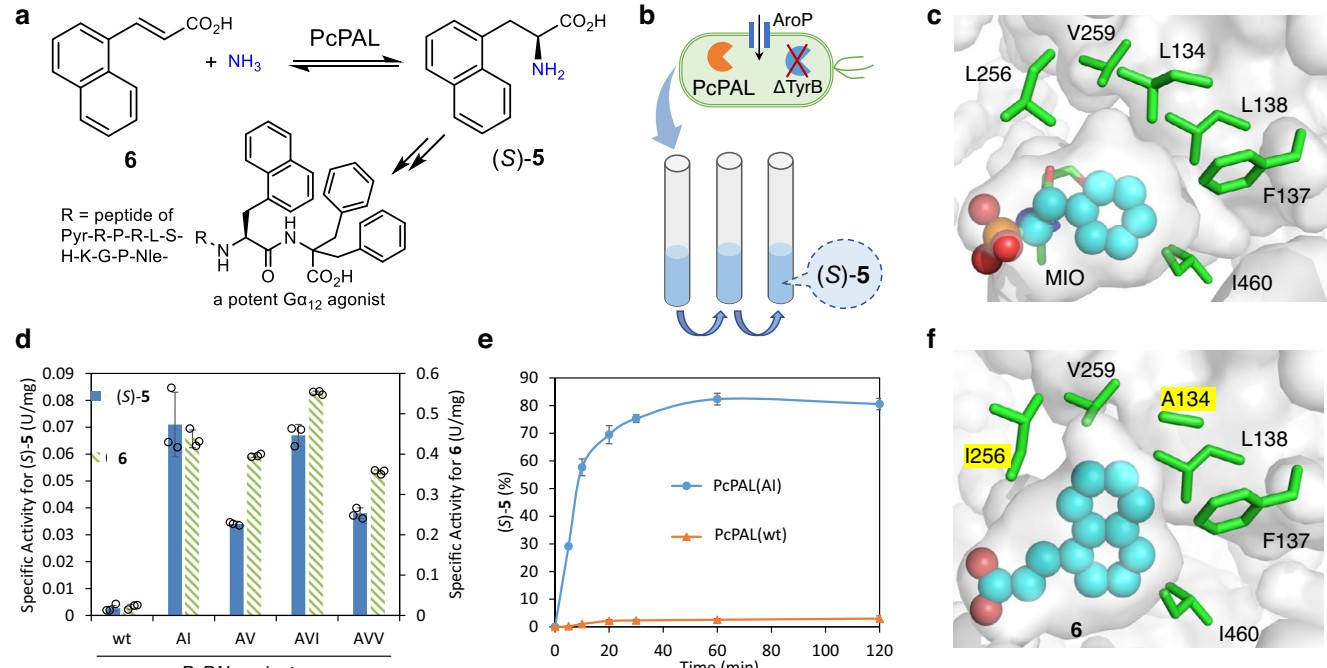

**Fig. 5 | Growth-based directed evolution of PcPAL for the asymmetric synthesis of (S)-5. a** Targeted reaction to convert **6** to (S)-**5**. **b** The principle of the growth selection is based on *E. coli* Δ*tyrB* co-expressing PcPAL variants and a transporter (AroP) in the M9 liquid medium where the amino acid (S)-**5** serves as the only nitrogen source. **c** Identification of six key residues by analyzing the active site of PcPAL(wt) (PDB: 6HQF) with (R)-(1-amino-2-phenylethyl)phosphonic acid (inhibitor, represented as cyan spheres). **d** Specific activities of purified PcPAL variants for converting (S)-**5** and **6**. **e** Time course of the asymmetric synthesis of (S)-**5** from **6** by *E. coli* expressing PcPAL(AI) or PcPAL(wt). **f** Docking of (S)-**5** (represented as cyan spheres) in the active site of the structure model of PcPAL(AI) with L134A/L256I. PcPAL: phenylalanine ammonia lyase from *Petroselinum crispum*; AroP: aromatic amino acid transporter from *E. coli*; TyrB: aromatic amino acid transaminase from *E. coli*; MIO: 4-methylideneimidazol-5-one. Source data are provided as a Source Data file. Data in **d** and **e** are mean values of triplicate experiments with error bars indicating the s.d. (*n* = 3).

growth on (S)-**5** was observed, deleting *tyrB* and co-expressing AroP may help to identify PcPAL variants with improved activity on (S)-**5**. Thus, the selection was performed using this engineered *E. coli* host in liquid cultures with 1 mM of (S)-**5** (Fig. 5b).

Taking advantage of previous studies of PcPAL[19], we aimed to improve its activity via a single round of evolution. Guided by the structure of PcPAL(wt) with the inhibitor (R)-(1-amino-2-phenylethyl) phosphonic acid (PDB: 6HQF)[65], six hydrophobic residues lining the phenyl ring (L134, F137, L138, L256, V259, I460) were selected for simultaneous multi-site mutagenesis (Fig. 5c). Instead of using NNK or DBS codons, we only focused on small hydrophobic side chains: the DYA codon (encodes L, I, V, T, S, and A) was used for positions L134, L138, and L256; the DYC codon (encodes F, I, V, T, S, and A) was used for F137; the RYC codon (encodes I, V, T, and A) was used for V259 and I460. Six-site directed mutageneses of PcPAL(wt) were achieved with sequentially performed Q5 mutagenesis rounds, leading to a library of 20,736 unique combinations. The libraries were constructed using the plasmids with strong to weak promoters, then introduced into *E. coli* BL21(DE3)Δ*tyrB* cells (expressing the transporter AroP) and subjected to enrichment in M9 medium supplemented with (S)-**5**. The culture was diluted every 24 h 5 times, and significant growth was only observed for the library with strong expression level (Supplementary Fig. 21). The resulting enriched culture was spread on a LB agar plate and 24 colonies were isolated to yield four unique variants: PcPAL(AI) with L134A/L256I, PcPAL(AV) with L134A/F137V, PcPAL(AVI) with L134A/F137V/L256I, and PcPAL(AVV) with L134A/F137V/I460V. These variants showed specific activities of 0.034–0.071 U/mg for converting (S)-**5** and specific activities of 0.36–0.55 U/mg for converting **6** (Fig. 5d). These activities are 13–26 times higher than that of PcPAL(wt). *E. coli* whole cells expressing PcPAL variants were evaluated for the asymmetric synthesis of (S)-**5** (Supplementary Fig. 22). The best

PcPAL(AI) afforded enantiopure (S)-**5** in 82% conversion in 1 h, whereas the wild-type PcPAL only gave (S)-**5** in 3% conversion under the same conditions (Fig. 5e, Supplementary Fig. 23).

To provide molecular insights for the improved activity, compound **6** was docked to the homology model of PcPAL(AI) (Fig. 5f). Clearly, **6** binds to the active site of PcPAL(AI) in the same pose as the inhibitor in the PcPAL(wt) (Fig. 5c) as shown from our docking models. The L134A mutation (most important, reflected in all variants) clearly caves out additional space in the pocket to accommodate the naphthyl group. The L256I mutation (found in the two most active variants) slightly and precisely enlarges the pocket to accommodate **6**. Another important mutation, F137V, also provides more space but in a direction that may not directly contact **6**. With these modifications, the specific activity of PcPAL towards **6** increased 26-fold to 0.55 U/mg, enabling the biocatalytic synthesis of the key chiral non-natural amino acid, though further engineering of PcPAL may be required for practical applications.

## Discussion

Fast identification of active enzyme mutants is crucial for the directed evolution of biocatalysts for the efficient synthesis of chemicals. Instead of the widely adopted high-throughput screening methods, herein, we developed a facile growth selection system – fine-tuned by a set of suitable promoters – generally applicable for different classes of amine-forming or converting enzymes. The principle of the methodology is based on the targeted chiral amine as the sole nitrogen source for *E. coli* cell growth in a chemically-defined medium. Our selection method is fast and efficient: for carrying out the protocol on agar plates, one week is sufficient to proceed from a cloned library to identified and sequenced hits. Two weeks are necessary if the selection is done via growth enrichment in liquid medium. A clear advantage of

this approach is that an organic phase can be used as a substrate/product reservoir if a hydrophobic substrate (or product) has to be used that shows a low solubility or toxicity at a higher concentration. The largest library in this study included $10^5$ clones for the selection of a MAO variant, keeping in mind an electroporation efficiency (= throughput) of $10^5$–$10^6$ in our study (Supplementary Fig. 24). The upper limit is only given by the transformation efficiency and might be increased to $10^6$–$10^7$ variants. This makes this strategy especially appealing for simultaneous site-saturated mutation of multiple positions to harness potentially synergistic effects and thus to identify variants with a significant improvement in one round of evolution. A potentially higher activity gain on the other hand also facilitates a clear growth advantage and the reduction of false-positive hits.

In comparison with previously developed screening methods using low/medium-throughput analytic instruments or high-throughput microfluidic devices, our growth selection method presented here can interrogate very large enzyme libraries by circumventing the analysis of every variant. Furthermore, the method works also without expensive and specialized instruments/devices. This is particularly useful for laboratories with limited access to these instruments/devices, such as small companies or labs in developing countries. In comparison with traditional screening in microtiter plates, the growth selection system is free from tedious work and substantially reduces the use of expensive consumables (such as by using MTP assays and/or robotic platforms). Considering that many laboratory consumables are plastics, the growth selection method furthermore has a minimized environmental footprint. Another advantage of the growth selection method is its general applicability: these growth selection procedures could be easily transferable to evolve other classes of amine-converting enzymes. The use of fine-tuned promoters and expression systems enables applying this growth selection concept to other enzyme classes and growth-facilitating metabolites produced. This general applicability also reduces the barriers for researchers to adopt this method.

In conclusion, our growth selection system was demonstrated to enable significant improvement of catalytic activities (26–270-fold) as shown for an amine transaminase, a monoamine oxidase, and a lyase in only one or two rounds of evolution without expensive and specialized equipment. We are convinced that the concept and method described herein are generally applicable to improve other enzymes (e.g., amine dehydrogenases, imine reductases) for the synthesis of non-natural chiral amines, amino acids, amides, and other nitrogen-containing molecules. Considering its attractive features of simplicity, high-throughput, environmental friendliness and low-equipment dependency, we envision that the growth selection system can be widely adopted in academia and industry to develop biocatalysts suitable for various applications.

## Methods

### Growth selection of AtTA libraries for converting (R)-1
The AtTA libraries with strong and medium constitutive promoters were transformed into electrocompetent cells of *E. coli* BL21 (DE3) by using an electroporation cuvette (1 mm gap) and default *E. coli* settings (1.8 kV) for the MicroPulser Electroporator (Biorad). Immediately after the electroporation shock, prewarmed LB medium (2 ml, 37 °C) was added to the cuvette and the cells were cultured in an incubator for 2 h at 30 °C, 200 rpm. Then, a small portion of the cells (20 µl) was isolated to test the electroporation efficiency. Half of the cells were added into LB medium (5 ml) containing kanamycin (50 mg l$^{-1}$) for isolation of the plasmids from the libraries. The other half of the cells were centrifuged (5 min, 4000 × $g$), and the supernatant of the LB medium was removed. The cell pellets were resuspended in M9 medium (300 µl) without nitrogen source, and then spread on the agar plates of M9 medium containing kanamycin (25 mg l$^{-1}$) and (R)-1 (10 mM) as the sole

nitrogen source. The plates were kept in an incubator (30 °C) for 5 days. Photos of the plates were taken every 24 h. Colonies were usually observed on the 2nd or 3rd day, if there are positive hits. Some big and representative colonies were picked and transferred into LB liquid medium (5 ml) containing kanamycin (50 mg l$^{-1}$) for preparation of cell stocks and plasmids for sequencing. The growth selection for the second round of evolution of AtTA was performed following the same procedures above.

### Asymmetric synthesis of (R)-1 by purified AtTA variants
The reaction was run in a vessel fitted with a magnetic stirrer, temperature probe, pH probe, and base addition. To a 100-ml flask, the following components were added to form a 40-ml system: CHES buffer (100 mM, pH 8.5) including PLP (1 mM), D-alanine (500 mM), oxidized nicotinamide adenine dinucleotide (NAD$^+$, 2–3 mM), glucose dehydrogenase (3 U ml$^{-1}$), D-glucose (550 mM) and lactate dehydrogenase (6 U ml$^{-1}$); 8 ml stock solution of **2** (1250 mM in DMSO); purified protein (1–2 mg ml$^{-1}$). The addition of the respective ketone started the reaction and the reactor was then stirred at 30 °C and with the pH maintained between 8.4–8.6 for 12 h by automatic addition of NaOH solution (2 M). To monitor the progression of the reaction, samples (50 µl) were taken at different times of the reaction (0.5, 1, 1.5, 2, 2.5, 3, 5, and 12 h), and mixed with acetonitrile (500 µl) and hydrochloric acid solution (100 mM, 450 µl) for HPLC analysis to determine the conversion of **2** (calibration curves, see Supplementary Fig. 25). Afterwards, the reaction was quenched with HCl to pH 2.0. The products were extracted with ethyl acetate (EtOAc) at pH 10 (NaOH addition), delivering (after drying and evaporation) the crude amines as oils. These were characterized (HPLC and achiral GC). The *ee* values of (R)-**1** were analyzed by chiral GC analysis. After extraction of (R)-**1** with EtOAc, the compound was dissolved in CH$_2$Cl$_2$, then derivatization to the trifluoroacetamide was performed by adding a 20-fold excess of trifluoroacetic anhydride. After purging with nitrogen to remove excess anhydride and residual trifluoroacetic acid, the derivatized compound was dissolved in CH$_2$Cl$_2$ and analyzed.

### Growth selection of CHAO libraries for converting (S)-3
The CHAO libraries with medium, weak, and very weak constitutive promoters were transformed into electrocompetent cells of *E. coli* BL21 (DE3) containing the plasmid SCm-KatE (constitutively expressing the catalase from *E. coli*) by the same electroporation procedure as described above. A small portion of the cells (20 µl) was used to test the electroporation efficiency and half of the cells were used to prepare the plasmids of the libraries. The other half of the cells were centrifuged (5 min, 4000 × $g$), and the supernatant of the LB medium was removed. The cell pellets were resuspended in M9 liquid medium (2 ml) containing appropriate antibiotics (25 mg l$^{-1}$ kanamycin and 12.5 mg l$^{-1}$ ampicillin) and (S)-**3** (1 mM) as the sole nitrogen source. Methyl laurate (0.5 ml) was added into the aqueous M9 medium as a growth-compatible second phase. The two-phase cultures were kept in an incubator (30 °C, 200 rpm), and diluted (10×) in fresh M9 medium every 24 h 5 times (5 days in total). The dilution was performed by adding existing aqueous culture (200 µl) into fresh M9 medium (1.8 ml) containing appropriate antibiotics (25 mg l$^{-1}$ kanamycin and 12.5 mg l$^{-1}$ ampicillin) and (S)-**3** (1 mM) and fresh methyl laurate (0.5 ml). Photos of the cultures were taken and the optical densities were measured every 24 h. An aliquot (50 µl) of the enriched culture of the CHAO library with medium expressing level was spread on an LB agar plate containing kanamycin (50 mg l$^{-1}$) for isolation of single colonies (37 °C, for 24 h). The representative colonies were picked and transferred into LB liquid medium (5 ml) containing kanamycin (50 mg l$^{-1}$) for preparation of cell stocks and plasmids for sequencing.

## Deracemization of *rac*-3 by CHAOs, catalase, and NH₃·BH₃

Fresh *E. coli* cells containing the vectors pRSF-CHAO(MVLM) or pRSF-CHAO(wt) and SCm-KatE were employed as whole-cell catalysts for the reaction. The cell pellets of a 50-ml culture were first washed with Tris-HCl buffer (100 mM, pH 8.0) and subjected to centrifugation again (4000 × *g*, 15 min). The supernatant was discarded and the cell pellets were resuspended in new Tris-HCl buffer (100 mM, pH 8.0) and the optical density of the cell suspensions was measured. To a reaction vial (20 ml) with screwcap, a stock solution of *rac*-3 (500 mM HCl salt in water), a stock solution of NH₃·BH₃ (1 M in Tris-HCl buffer), Tris-HCl buffer (100 mM, pH 8.0), and *n*-dodecane (1 ml) were added to form a two-phase catalytic system composed of the aqueous buffer (1 ml) containing *rac*-3 (25 mM), NH₃·BH₃ (500 mM) and the organic phase (1 ml). The cell suspensions were added to start the reaction (final density of cells in aqueous phase was 10 g l⁻¹ dcw, dry cell weight). Thirty-six reaction vials were incubated at 30 °C, 200 rpm for 4 h (the caps were opened for venting at 1 and 2 h for 1 min). To monitor the progress of the reaction, NaOH solution (5 M, 100 μl) was added to the reaction vials at different times (0, 0.25, 0.5, 1, 2, and 4 h) to stop the reaction. Then, *n*-hexane (1 ml) was added to the vial for extraction. The organic phase (*n*-hexane or *n*-dodecane) was centrifuged (12,000 × *g*, 3 min) and an aliquot of the supernatant (100 μl) was mixed with *n*-hexane (900 μl, containing 2 mM ethylbenzene as internal standard) for simultaneous chiral HPLC analysis of the conversion and *ee*.

## Growth selection of PcPAL libraries for converting (*S*)-5

The PcPAL libraries with strong, medium, and weak constitutive promoters were transformed into electrocompetent cells of *E. coli* BL21 (DE3) Δ*tyrB* containing the plasmid SCm-AroP (constitutively expressing the aromatic amino acid transporter from *E. coli*) by the same electroporation procedure as described above. A small portion of the cells (20 μl) was used to test the electroporation efficiency and half of the cells were used to prepare the plasmids of the libraries. The other half of the cells were centrifuged (5 min, 4000 g), and the supernatant of the LB medium was removed. The cell pellets were resuspended in M9 liquid medium (2 ml) containing appropriate antibiotics (25 mg l⁻¹ kanamycin and 12.5 mg l⁻¹ ampicillin) and (*S*)-5 (1 mM) as the sole nitrogen source. The cultures were kept in an incubator (30 °C, 200 rpm), and diluted (10x) in fresh M9 medium every 24 h 5 times (5 days in total). The dilution was performed by adding the existing aqueous culture (200 μl) into fresh M9 medium (1.8 ml) containing appropriate antibiotics (25 mg l⁻¹ kanamycin and 12.5 mg l⁻¹ ampicillin) and (*S*)-5 (1 mM). The optical densities of the cultures were measured for the last three days, and a photo was taken at the end of enrichment. The enriched culture of PcPAL library with strong expressing level was spread on an LB agar plate containing kanamycin (50 mg l⁻¹) for isolation of single colonies (37 °C, for 24 h). The representative colonies were picked and transferred into LB liquid medium (5 ml) containing kanamycin (50 mg l⁻¹) for preparation of cell stocks and plasmids for sequencing.

## Asymmetric synthesis of (*S*)-5 by PcPAL variants

Fresh *E. coli* cells containing the vectors pRSF-PcPAL(AI) or pRSF-PcPAL(wt) were employed as whole-cell catalysts for the reaction. The cell pellets from a 50-ml culture were resuspended in NH₃/NH₄Cl buffer (6 M, pH 10.0) and the optical density of the cell suspensions was measured. To a reaction vial (20 ml) with a screwcap, a stock solution of **6** (1 M in DMSO) and NH₃/NH₄Cl buffer (6 M, pH 10.0) were added to form a catalytic system (1 ml) containing **6** (22 mM). The cell suspensions were added to start the reaction (final density of cells was 5 g dcw l⁻¹). Forty-two reaction vials were incubated at 30 °C, 200 rpm for 2 h. To monitor the progress of the reaction, HCl solution (6 M, 1 ml) was added to the reaction vials at different times (0, 5, 10, 20, 30, 60, and 120 min) to stop the reaction. To determine the conversion, methanol (2 ml, containing 2 mM acetophenone as the internal

standard) was added to the vial. An aliquot of the mixture (1 ml) was centrifuged (12,000 × *g*, 10 min), and an aliquot of the supernatant (700 μl) was used for HPLC analysis of the conversion. To determine the *ee* of **5**, methanol (300 μl) was added to the vial. The mixture was centrifuged (12,000 × *g*, 10 min), and an aliquot of the supernatant (700 μl) was used for chiral HPLC analysis.

## Other related methods

Primers information is provided in the Supplementary Table 2. Other related methods are provided as Supplementary Methods in the Supplementary Information file.

## Reporting summary

Further information on research design is available in the Nature Portfolio Reporting Summary linked to this article.

## Data availability

The structural data used in this study are from Protein Data Bank under accession codes 4CE5, 4I59, and 6HQF. The structural data generated in this study have been deposited in the Protein Data Bank under accession codes 7XG5 and 7XG6. Additional data supporting the findings of this study are available as Supplementary Information. All unique biological materials (plasmids and strains) are readily available from the corresponding authors upon request. Source data are provided with this paper.

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

## Acknowledgements

We acknowledge the financial supports from the National Natural Science Foundation of China (No. 32101229 to S.W.), the Alexander von Humboldt-Stiftung (to S.W.), and the China Scholarship Council (a Ph.D. thesis project to C.X., File No.: 201808330394). W.L. thanks the financial supports from Tianjin Synthetic Biotechnology Innovation Capacity Improvement Project (TSBICIP-PTJJ-008, TSBICIP-IJCP-003, TSBICIP-KJGG-009-01, TSBICIP-KJGG-002-06), Youth Innovation Promotion Association CAS and China Scholarship Council. We thank Martin Woicke for preliminary experiments on CHAO, Ina Menyes for support with chiral GC analysis, Prof. Michael Lammers for providing reagents and facilities for crystallization, Prof. Huai-Long Teng for support with chiral HPLC analysis, and the staff from the Beamline 14.1 at BESSY for assistance during data collection.

## Author contributions

S.W. and U.T.B. initiated the project. S.W. designed the study and selected the enzymes and targets. C.X. performed the evolution of AtTA; S.W. and M.S.H.K. performed the evolution of CHAO; S.W. and Y.Z. performed the evolution of PcPAL. W.L., C.G.F., R.W. and G.W. solved the crystal structure of AtTA(RHC). S.W., C.X., M.H. and U.T.B. discussed the findings and wrote the manuscript with inputs from all authors.

## Funding

## Competing interests

The authors declare no competing interests.
