## [Peer Review File · Nature Communications]

A growth selection system for the directed evolution of amine-forming or converting enzymesEditorial Note: This manuscript has been previously reviewed at another journal that is not operating a transparent peer review scheme. This document only contains reviewer comments and rebuttal letters for versions considered at Nature Communications.

REVIEWER COMMENTS

Reviewer #2 (Remarks to the Author):

The authors have addressed my points in a satisfactory manner. However there is still some inconsistency that I perhaps should have noticed the first time:

In Supplementary Table 1, the structure deposited in the PDB with ID 7XG5 is described as a PLP complex and 7XG6 as an LLP complex. On p. 10 of the manuscript, 7XG5 is described as an apo form and 7XG6 as a PLP complex. In Supplementary Figure 12, 7XG5 is described as a complex with both PLP and (R)-1 and the (R)-1 part is shown in the figure, though its fit to the density is not great. Also, on p. 42 of the Supplementary Material it is written that "The existing PLP in the crystal structure of AtTA(RHC) (PDB: 7XG5) was removed to prepare the structure for docking.", again implying that only PLP was present. Is the deposited structure with or without (R)-1, i.e. what is actually in the deposited file? In any case, the statements within this manuscript and its supplementary material should be made consistent.

Reviewer #3 (Remarks to the Author):

the authors have addressed all my comments and I only have minor comments raised when reading the answers of the authors.

Regarding the catalase sequence, when the authors say that there are no mutations introduced, do they mean that they sequenced the gene after the evolution process and did not find any mutation? this was not specifically mentioned or shown with seq data.

On the repetition of Supp fig 22, it seems to be a bit of discrepancy with some of the results obtained in the previous version of the figure (without replicates). Where is the discrepancy coming from?

Reviewer #4 (Remarks to the Author):

The authors have now made a number of significant revisions to the original manuscript, which was submitted to a sister journal. Specifically they have addressed some of the concerns of this review with respect to (i) including additional references to prior biocatalytic routes to the target molecules and (ii) including a new paragraph which discusses the relative merits of the screening method reported relative to other approaches. They have also addressed issues raised by the other reviewers. The manuscript in my opinion is now suitable for publication.

Reviewer #5 (Remarks to the Author):

The manuscript describes 3 instances of growth-based microbial selections of "small but smart" combinatorial site-saturation mutagenesis libraries of 3 different ammonia- or alanine-

releasing enzymes, relevant for the preparation of chiral amines. The 3 case studies showcase the peculiarities of each enzyme (toxic and/or insoluble substrates, lower and higher starting activities, etc.). Through the differences in these 3 case studies, the authors illustrate how to circumvent a range of common pitfalls of selections that readers may find when tailoring the selection presented here to their particular systems. This is a very strong aspect of the manuscript.

Although nutritional selections are not new, they have been greatly overlooked for directed evolution purposes, due to their limitation to specific molecules and enzymes. This work fills the void by linking the conversion of complex molecules relevant for biocatalysis to simple nutrients, such as ammonia or alanine. The simplicity of the approach is inherent to the methodology used but nevertheless much appreciated, especially considering the technical complexity of alternatives such as UPLC or droplet microfluidics.

Responses to reviewer 1:

Comment 1:

The comment about novelty in the specific application of selection of improved enzymes for non-natural chiral amines has been answered satisfactorily, as the authors successfully demonstrate the synthesis of 3 amines and the authors suggest in the Discussion (line 336) the general idea that ANY amine-forming/converting enzyme (not just chiral amine-forming enzymes) releasing a key macronutrient even from the most complex, “non-natural” substrates can be selected for, as long as the pre-requisites listed by the authors are met (lines 116-111).

Comment 2:

In my opinion, the authors have replied to the comment satisfactorily.

Comment 3:

The authors have replied to the comment satisfactorily. However, not enough emphasis is made again on the positive qualities of the method. For instance, the authors’ rationale that a directed evolution campaign involves tedious work and high use of consumables could be better appreciated by readers if the authors stated the real throughput of their method. Unfortunately, I could not find this number in this manuscript, which can be easily calculated by plating their libraries under non-selective conditions. Furthermore, this number could also be used to calculate the oversampling and the selection factor in the different selections. This could support some of the authors’ arguments more quantitatively, e.g. on the capacity of the selection method to handle larger libraries or on stringency. Please, calculate these numbers and introduce them in an adequate place in the manuscript.

Responses to reviewer 2:

All questions:

In my opinion, the authors have replied to the questions satisfactorily and made the pertinent modifications in the manuscript.

Responses to reviewer 3:

In my opinion, the authors have replied to the questions satisfactorily and made pertinent modifications where required. However, some of the ideas provided in answers should be

incorporated in the manuscript, to clarify their rationale to the reader or, again, to emphasize the relevance of their clever approach and good results. Specifically:

- the fact that these enzyme variants are the best created so far (to the best of their knowledge)
- the rationale behind using crystal structures for the AtTA variant but models for CHAO and PcPAL.

In addition, I agree with the authors' statement that Fig S20 (there is a misnumbering in their reply) shows that when the background problem is fixed by the deletion of *tyrB*, there seems to be a better correlation between growth and promoter strength. For the sake of precision, I suggest that the authors rephrase their added sentence from "growth depends on the expression level of PcPAL" to "there was a better correlation between growth and promoter strength".

Responses to reviewer 4:

In my opinion, the authors have replied to the questions satisfactorily and made pertinent modifications where required.

However, the authors should quantitatively demonstrate that their method can be described as "ultra high-throughput", as they state in the manuscript and in their reply to Reviewer 4. The term "ultra high-throughput" is commonly used in the community to signify a throughput of or equivalent to that of FACS (i.e. 10000 events/second). This means, that 36 million individuals can be processed in an hour. In this work, the authors select an unknown number of individuals (see my comment above about estimation of library size) in 2-5 days.

Point-by-Point Response to the Reviewers' Comments of NCOMMS-22-24818-T

REVIEWER COMMENTS

Reviewer #2 (Remarks to the Author):

The authors have addressed my points in a satisfactory manner. However there is still some inconsistency that I perhaps should have noticed the first time:

In Supplementary Table 1, the structure deposited in the PDB with ID 7XG5 is described as a PLP complex and 7XG6 as an LLP complex. On p. 10 of the manuscript, 7XG5 is described as an apo form and 7XG6 as a PLP complex. In Supplementary Figure 12, 7XG5 is described as a complex with both PLP and (R)-1 and the (R)-1 part is shown in the figure, though its fit to the density is not great. Also, on p. 42 of the Supplementary Material it is written that "The existing PLP in the crystal structure of AtTA(RHC) (PDB: 7XG5) was removed to prepare the structure for docking.", again implying that only PLP was present. Is the deposited structure with or without (R)-1, i.e. what is actually in the deposited file? In any case, the statements within this manuscript and its supplementary material should be made consistent.

Response: We are grateful for the positive evaluation of our revised manuscript and apologize for the confusion as pointed out by reviewer#2.

The PDB file 7XG5 is a PLP complex (with a partial density of (R)-1)) and 7XG6 is an LLP complex. To keep consistency, we corrected the main text on p. 10 to: "we solved the crystal structure of AtTA(RHC) (Supplementary Fig. 11, Supplementary Table 1) with bound PLP (7XG5) and with bound LLP (PLP-Lys180, 7XG6)". We corrected the legend of Supplementary Figure 12a to: "a, 2m|Fo|-D|Fc| map (contoured at 1.0 σ) of the AtTA(RHC) structure (PDB: 7XG5) with PLP (and a partial density was observed for (R)-1)." We also corrected the method given in the SI p. 42 to: "The existing PLP and (R)-1 (partial density) in the crystal structure of AtTA(RHC) (PDB: 7XG5) was removed to prepare the structure for docking."

Reviewer #3 (Remarks to the Author):

The authors have addressed all my comments and I only have minor comments raised when reading the answers of the authors.

Regarding the catalase sequence, when the authors say that there are no mutations introduced, do they mean that they sequenced the gene after the evolution process and did not find any mutation? this was not specifically mentioned or shown with seq data.

Response: Thanks a lot for this comment. During the evolution process, we only sequenced the targeted enzyme (CHAO) but did not sequence the assisting enzyme (catalase) because of the following two reasons. 1) We did not introduce mutations into the catalase and it is also unlikely to have spontaneous mutations during the 5 days' of cultivation. 2) According to the Supplementary Figures 13 and 14, the limitation for the growth is the activity of CHAO. Thus, it is unlikely to have a selection pressure on the assisting catalase, which already has a very high specific activity (11.7 kU/mg) for degrading H₂O₂ (Eur. J. Biochem, 1995, 230, 127). Therefore, we only sequenced the targeted CHAO after the evolution process.

On the repetition of Supp fig 22, it seems to be a bit of discrepancy with some of the results obtained in the previous version of the figure (without replicates). Where is the discrepancy coming from?

Response: Thanks a lot for this comment. For the repetition of Supp Fig 22, the reaction conditions are slightly different from those given in the previous manuscript version. The previous figure is the result of biocatalysis with *E. coli* cells (10 g dcw l⁻¹) for 24 h, while the repetition shown in Supp Fig 22 is the result of biocatalysis with *E. coli* cells (2 g dcw l⁻¹) for 6 h. We think this could be the cause for the minor discrepancy of the results. Nevertheless, the main purpose of Supp Fig 22 is to identify the best mutant to synthesize (S)-5, and the conclusion is still that PcPAL(AI) is the best enzyme. This is consistent with the previous version of the figure.

Reviewer #4 (Remarks to the Author):

The authors have now made a number of significant revisions to the original manuscript, which was submitted to a sister journal. Specifically they have addressed some of the concerns of this review with respect to (i) including additional references to prior biocatalytic routes to the target molecules and (ii) including a new paragraph which discusses the relative merits of the screening method reported relative to other approaches. They have also addressed issues raised by the other reviewers. The manuscript in my opinion is now suitable for publication.

Response: We are very grateful that reviewer#4 considers our manuscript suitable for publication.

Reviewer #5 (Remarks to the Author):

The manuscript describes 3 instances of growth-based microbial selections of “small but smart” combinatorial site-saturation mutagenesis libraries of 3 different ammonia- or alanine-releasing enzymes, relevant for the preparation of chiral amines. The 3 case studies showcase the peculiarities of each enzyme (toxic and/or insoluble substrates, lower and higher starting activities, etc.). Through the differences in these 3 case studies, the authors illustrate how to circumvent a range of common pitfalls of selections that readers may find when tailoring the selection presented here to their particular systems. This is a very strong aspect of the manuscript.

Although nutritional selections are not new, they have been greatly overlooked for directed evolution purposes, due to their limitation to specific molecules and enzymes. This work fills the void by linking the conversion of complex molecules relevant for biocatalysis to simple nutrients, such as ammonia or alanine. The simplicity of the approach is inherent to the methodology used but nevertheless much appreciated, especially considering the technical complexity of alternatives such as UPLC or droplet microfluidics.

Response: We are very grateful for the overall positive evaluation of our manuscript and very much appreciate the time and effort dedicated by reviewer#5 especially as he/she had not seen the initial submission.

Responses to reviewer 1:

Comment 1:

The comment about novelty in the specific application of selection of improved enzymes for non-natural chiral amines has been answered satisfactorily, as the authors successfully demonstrate the synthesis of 3 amines and the authors suggest in the Discussion (line 336) the general idea that ANY amine-forming/converting enzyme (not just chiral amine-forming enzymes) releasing a key macronutrient even from the most complex, “non-natural” substrates can be selected for, as long as

the pre-requisites listed by the authors are met (lines 116-111).

Response: Thanks a lot for this valuable comment.

Comment 2:

In my opinion, the authors have replied to the comment satisfactorily.

Response: Thanks a lot for this statement.

Comment 3:

The authors have replied to the comment satisfactorily. However, not enough emphasis is made again on the positive qualities of the method. For instance, the authors' rationale that a directed evolution campaign involves tedious work and high use of consumables could be better appreciated by readers if the authors stated the real throughput of their method. Unfortunately, I could not find this number in this manuscript, which can be easily calculated by plating their libraries under non-selective conditions. Furthermore, this number could also be used to calculate the oversampling and the selection factor in the different selections. This could support some of the authors' arguments more quantitatively, e.g. on the capacity of the selection method to handle larger libraries or on stringency. Please, calculate these numbers and introduce them in an adequate place in the manuscript.

Response: Thanks for the comments. We have tested the electroporation efficiency when performing the growth selection-based evolution. The electroporation efficiency (= throughput) was usually 10^5 - 10^6 in our study, and we added a sentence in the discussion part to describe this and provided a new Supplementary Fig. 24 (copied below).

Supplementary Figure 24. Typical results of testing the electroporation efficiency by plating a small portion (1/1000) of transformed cells on LB-agar plates. The electroporation efficiency was usually 10^5 - 10^6 in our study.

To further emphasize the positive quality of the method, we added two sentences about the environmental effect in the discussion part. "In comparison with traditional screening in microtiter plates, the growth selection method is free from tedious work and substantially reduces the use of expensive consumables (such as by using MTP assays and/or robotic platforms). Considering that many laboratory consumables are plastics, the growth selection method furthermore has a minimized environmental footprint." We also added the merit "environmentally friendly" in the conclusion paragraph.

Responses to reviewer 2:

All questions:

In my opinion, the authors have replied to the questions satisfactorily and made the pertinent modifications in the manuscript.

Response: Thanks for the comments.

Responses to reviewer 3:

In my opinion, the authors have replied to the questions satisfactorily and made pertinent modifications where required. However, some of the ideas provided in answers should be incorporated in the manuscript, to clarify their rationale to the reader or, again, to emphasize the relevance of their clever approach and good results. Specifically:

- the fact that these enzyme variants are the best created so far (to the best of their knowledge)
- the rationale behind using crystal structures for the AtTA variant but models for CHAO and PcPAL.

Response: As suggested, we added two sentences to emphasize the superiority of the enzyme variants “To our knowledge, AtTA(RHC) is the best TA with known sequence for the synthesis of (R)-1.” and “As far as we know, CHAO(MVLM) is the best enzyme for the production of (R)-3 by deracemization.” We also added a sentence to explain the rationale for obtaining the crystal structure of AtTA(RHC) “(considering the changes in the charge, polarity, and size present in the triple mutant H55R/F115H/E117C)”.

In addition, I agree with the authors’ statement that Fig S20 (there is a misnumbering in their reply) shows that when the background problem is fixed by the deletion of tyrB, there seems to be a better correlation between growth and promoter strength. For the sake of precision, I suggest that the authors rephrase their added sentence from “growth depends on the expression level of PcPAL” to “there was a better correlation between growth and promoter strength”.

Response: As suggested, we rephrased the sentence to “there was a better correlation between growth and promoter strength”.

Responses to reviewer 4:

In my opinion, the authors have replied to the questions satisfactorily and made pertinent modifications where required.

However, the authors should quantitatively demonstrate that their method can be described as “ultra high-throughput”, as they state in the manuscript and in their reply to Reviewer 4. The term “ultra high-throughput” is commonly used in the community to signify a throughput of or equivalent to that of FACS (i.e. 10000 events/second). This means, that 36 million individuals can be processed in an hour. In this work, the authors select an unknown number of individuals (see my comment above about estimation of library size) in 2-5 days.

Response: As the electroporation efficiency is about 10^5 - 10^6 (though theoretically can be up to 10^7), the throughput of our method is 10^5 - 10^6 individuals in 2-5 days. This throughput is lower than that of FACS (3.6×10^7 individuals in 1 h). Thus, we corrected the “ultra high-throughput” to “high-throughput” to avoid an overstatement of the throughput of our method.

REVIEWERS' COMMENTS

Reviewer #2 (Remarks to the Author):

The authors have answered my remaining comments satisfactorily and the manuscript is suitable for publication.

Reviewer #3 (Remarks to the Author):

The authors answered my comments and I don't have further comments

Reviewer #5 (Remarks to the Author):

Thank you for considering my comments and answering my questions. All the pending items have now been satisfactorily solved.